

# Sex differences in wrist strength: a systematic review

Alexis D. Napper[1], Meera K. Sayal[1], Michael W.R. Holmes[1] and Alan C. Cudlip[2]

[1] Department of Kinesiology, Brock University, St. Catharines, Ontario, Canada
[2] Department of Kinesiology and Health Sciences, University of Waterloo, Waterloo, Ontario, Canada

## ABSTRACT

Sex differences in strength have been attributed to differences in body anthropometrics and composition; these factors are often ignored when generating workplace guidelines. These differences directly impact the upper extremity, leaving female workers exposed to injury risk. The wide range of tools and techniques for measuring upper extremity strength presents a challenge to ergonomists and work task designers; collating outcomes to provide a clear outlook of differences between males and females is essential and the purpose of this work. Four online databases were searched (PROSPERO ID: CRD42022339023) with a focus on articles assessing sex differences in wrist strength. A total of 2,378 articles were screened for relevancy; 25 full-text articles were included in this systematic review. Articles examined movement pairs (ulnar/radial deviation, pronation/supination, and flexion/extension), as well as contraction types (isometric and isokinetic) to observe sex differences in wrist strength. Across all articles, females produced ~60–65% of male flexion/extension strength, ~55–60% pronation/supination strength, and ~60–70% ulnar/radial deviation strength. Overall, females presented lower strength-producing abilities than males, but when considering strength relative to body mass, male-female differences were less pronounced and occasionally females surpassed male strength metrics; typically, this occurred during flexion/extension, particularly in isokinetic contractions. This review has identified a scarcity of articles examining ulnar/radial deviation, pronation/supination, as well as isokinetic contractions; these are needed to supplement workplace exposure guidelines.

# INTRODUCTION

Sex differences in strength generation are multifactorial. In the context of human capacity, strength is the ability to generate maximal force. Sex differences in strength have been attributed to differences in body composition and anthropometrics (*Glenmark et al., 2004*). At the same body mass index (BMI), males tend to have more lean muscle mass while females have more body fat than males (*Bredella, 2017*; *Tyagi et al., 2017*). These sex differences are more apparent during puberty when the production of sex hormones is increased (*Bredella, 2017*; *Tyagi et al., 2017*). These differences are more visible in the upper extremity; in absolute values and relative to body mass males have significantly greater skeletal muscle mass than females (*Miller et al., 1993*; *Janssen et al., 2000*). When examining equivalently trained athletes, females typically produce 40–75% of male absolute strength

Corresponding author
Alan C. Cudlip,
alan.cudlip@uwaterloo.ca

capabilities (*Bartolomei et al., 2021*). The strength-anthropometric relationship supports the notion that greater anthropometric measurements such as height, weight, limb length, and cross-sectional area (CSA) lead to greater strength production. This is supported by a larger CSA composed of a greater number of motor units and muscle fibers, leading to an increase in the ability to produce force (*Bartolomei et al., 2021*; *Qazi et al., 2017*). This potential mismatch in strength capability can affect activities of daily living, sports settings, or in the workplace.

The prevalence of female workers in the global labour force has increased. In 2014, 47% of the US labour force were women, and these projections suggest that by the year 2050, the number of female workers in the United States will surpass 92 million (*Cruz Rios, Chong & Grau, 2017*). Despite the increase in female workers, there is a lack of understanding of the differences in workplace risks between sexes (*Tessier-Sherman et al., 2014*). In jobs that require physical strength, female workers are more susceptible to workplace musculoskeletal disorders (WMSDs) as job demands represent a higher proportion of physical capacity (*Tessier-Sherman et al., 2014*; *Blue, 1993*). As job demands are typically indiscriminate of the sex of the worker, females are left with a higher risk of WMSDs than male counterparts in typical blue-collar jobs (*Taiwo et al., 2008*). Females are more susceptible to upper extremity repetitive strain injuries (RSIs) due to their smaller stature, hormonal differences, and decreased strength production (*Taiwo et al., 2008*). This correlates to an increased risk for carpal tunnel syndrome (CTS) and tendinopathies due to repetitive strain and physical demand requirements, particularly in females (*Fan et al., 2009*; *McDiarmid et al., 2000*; *Shiri & Viikari-Juntura, 2011*; *Wolf et al., 2010*).

To prevent this increased risk of WMSDs in female workers, understanding differences in strength capabilities between sexes is essential. While differences in wrist strengths between sexes have been documented, due to the large variabilities in strength findings across literature, task design and ergonomic guidelines are often driven based on a singular or small number of former research studies, which may have dramatic effects on what is deemed "acceptable" or "unacceptable" for workplace strength requirements. Workplace tasks may not scale force or loading requirements to worker anthropometrics or capability, placing disproportionate risk levels on those with decreased capacity. As these differences often align with sex differences, incorporating this into industrial design would provide the ability to strengthen workplace guidelines regarding the distal upper extremity to maximize worker safety. There is a wide body of research examining sex differences and wrist strength, but interpretation is hampered by varied measurement techniques and participant populations; collation of this existing research is essential to allow informed decision making by researchers. The purpose of this review was to assess the current literature on sex differences in wrist strength and to combine these findings to quantify changes in strength-generating abilities.

## MATERIALS AND METHODS

A search was conducted considering the main topics and selecting a concise list of keywords used to effectively extract relevant articles for consideration. Four databases were searched

**Table 1  Search string for all databases.** Search string generated from specific search terms to gather relevant articles regarding forearm and wrist segment and strength outcomes for both sexes.

| General area | Specific search terms |
| --- | --- |
| Body segment | Wrist **OR** forearm |
| Measurement of strength | Muscle **OR** joint **OR** isometric **OR** isokinetic **OR** dynamic |
| | **AND** |
| Sex identification | Strength **OR** force **OR** moment **OR** torque ((Woman **OR** women **OR** female) **AND** (Man **OR** men **OR** male)) |
| | **OR** |
| | Sex **OR** gender |

on 13 June 2022 including Embase, MedLine, ProQuest, and Web of Science. The search was generated, and a finalized search string for each database was produced (Table 1). The search string was used to ensure relevant articles providing information regarding the forearm and wrist segment and strength outcomes for both sexes were discovered. The study was registered to PROSPERO (ID: CRD42022339023).

Articles were deemed eligible if they contained the desired study design that focused on sex differences in strength metrics at the wrist. Studies were required to include both sexes and were limited to healthy individuals between 18 and 65 with no prior acute pathologies of the upper extremity, or chronic pathologies such as cancer or post-stroke diagnoses that could have impacted muscular strength. Articles needed to include strength modalities, with outcome measures for participant populations split between sexes. Exclusion criteria included incorrect study design, results not split by sex, wrong outcome measures, or if the article was not written in English.

A multi-step screening process was completed to filter articles for final data extraction. Initial database searches were performed based on inclusion/exclusion criteria. Articles were uploaded into Covidence (Veritas Health Innovation Ltd., Melbourne, Australia) to organize the screening processes. Titles and abstracts were examined first by two independent reviewers, with a third reviewer to resolve conflicts. Following initial screening, the same inclusion/exclusion criteria and conflict resolution approach was used in full-text review for eligible articles (Fig. 1). All articles that were accepted after the second round of screening were included for data extraction.

During extraction, all included articles were assessed for potential risk of bias using the Risk of Bias in Non-randomized Studies (ROBINS-I). Seven domains used to determine the level of bias within each article, including bias due to confounding, selection of participants, clarification of interventions, deviations from intended interventions, missing data, measurement outcomes, and selection of the reported results. Ratings were described as low (L), moderate (M), and serious (S) and a total bias rating was generated based on the highest score across domains (Table 2).

Following screening, relevant outcome information was extracted. Principal summary measures extracted from all articles were recorded strength metrics for males and females in each experimental task. In addition, participant demographics, measurement method, movement direction and type were extracted. Where available, effect sizes of these strength
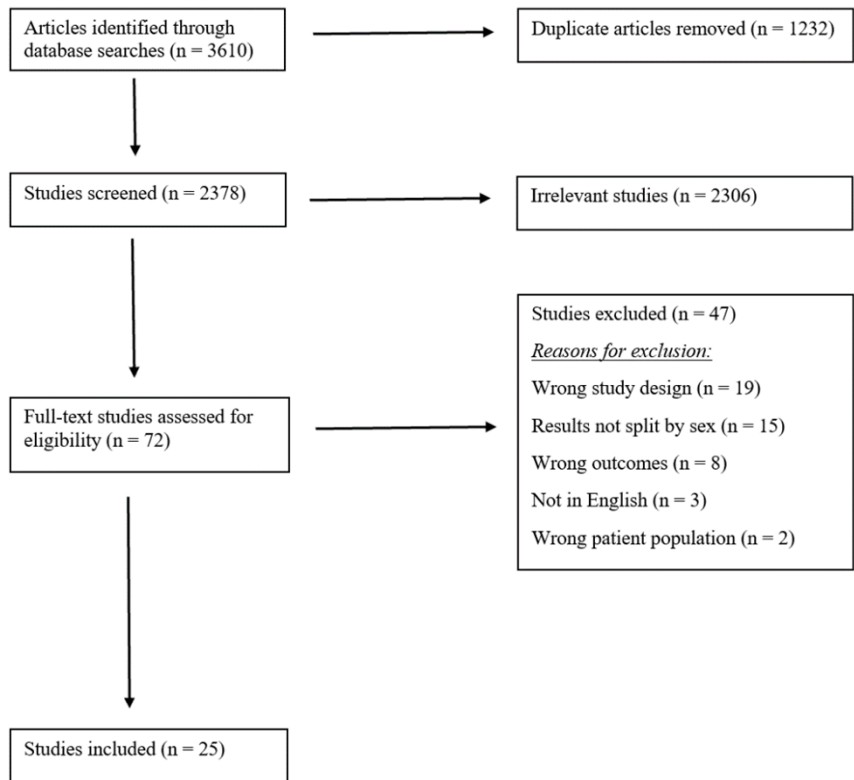

Figure 1 PRISMA flowchart describing screening process and inclusion of eligible articles for this review.

differences were recorded or calculated by using means and standard deviations between male and female groups and was included within the extraction tables.

## RESULTS

### Article selection

Collectively, 3,610 articles were collected through the initial database search; after the first screening stage and removal of duplicates, 2,306 articles were deemed irrelevant and 72 were full-text assessed for eligibility. Of these 72 articles, 47 were removed due to exclusion criteria, leaving 25 articles for extraction. A detailed flowchart of article screening is detailed in Fig. 1.

### Participant pool

Across articles, male and female participant age ranges were between 18-65. The mean age for males and females was 30.0 and 30.8, respectively. Nine articles included equal sample sizes of both males and females (*Andrews, Thomas & Bohannon, 1996*; *Crawford, Wanibe & Nayak, 2002*; *Forthomme et al., 2002*; *Hallbeck, 1994*; *Hill et al., 2018*; *Holzbaur et al., 2007*; *La Delfa et al., 2015*; *Plewa, Potvin & Dickey, 2016*; *Puharic & Bohannon, 1993*).

**Table 2 Risk of bias assessments using the ROBINS-I tool.**

| Article | 1 | 2 | 3 | 4 | 5 | 6 | 7 | Total |
|---|---|---|---|---|---|---|---|---|
| *Andrews, Thomas & Bohannon (1996)* | L | M | L | L | L | L | L | L |
| *Axelsson et al. (2018)* | M | L | L | L | L | L | L | M |
| *Cornu, Maietti & Ledoux (2003)* | L | L | L | L | L | L | L | L |
| *Cornwall (1994)* | L | L | L | L | L | L | L | L |
| *Crawford, Wanibe & Nayak (2002)* | L | L | L | L | L | L | L | L |
| *Danneskiold-Samsøe et al. (2009)* | L | M | L | L | L | L | L | M |
| *Forthomme et al. (2002)* | L | L | L | L | L | L | L | L |
| *Hallbeck (1994)* | L | L | L | L | L | L | L | L |
| *Harbin, Leyh & Harbin (2020)* | L | M | L | L | L | L | L | M |
| *Harbo, Brincks & Andersen (2012)* | L | M | L | L | M | M | L | M |
| *Hill et al. (2018)* | L | L | L | L | M | M | M | M |
| *Holzbaur et al. (2007)* | L | M | L | L | L | L | L | M |
| *Karahan et al. (2017)* | L | M | L | L | L | L | L | M |
| *Kramer et al. (1993)* | L | L | L | L | L | L | L | L |
| *La Delfa et al. (2015)* | L | M | L | L | M | L | L | M |
| *Lorbergs et al. (2011)* | L | L | L | L | L | L | L | L |
| *Mao et al. (2000)* | L | M | L | L | L | L | L | M |
| *Matsuoka et al. (2006)* | L | L | L | L | L | M | L | M |
| *Miller, Nair & Baratz (2005)* | L | L | L | L | L | L | L | L |
| *Nicholas et al. (1989)* | L | M | L | L | S | M | L | S |
| *Nilsson et al. (2019)* | L | L | L | L | L | L | L | L |
| *Plewa, Potvin & Dickey (2016)* | L | L | L | L | L | L | L | L |
| *Puharic & Bohannon (1993)* | L | L | L | L | L | L | L | L |
| *Raschner et al. (2010)* | L | M | L | L | L | L | L | M |
| *Richards, Gordon & Beaton (1993)* | L | L | L | M | L | L | L | M |

**Notes.**
Bias Domains: (1) Bias due to confounding; (2) bias in the selection of participants to the study; (3) bias in the classification of interventions; (4) bias due to deviations from intended interventions; (5) bias due to missing data; (6) bias in measurement outcomes; (7) bias in the selection of the reported result. The total score is the highest risk value across domains.
L, low; M, moderate; S, serious risk of bias.

## Contraction type and movement direction

Multiple movements and contraction types were examined. Movement directions were organized into three pairs: flexion/extension ($n = 12$; *Cornu, Maietti & Ledoux, 2003*; *Danneskiold-Samsøe et al., 2009*; *Forthomme et al., 2002*; *Hallbeck, 1994*; *Harbo, Brincks & Andersen, 2012*; *Holzbaur et al., 2007*; *Karahan et al., 2017*; *La Delfa et al., 2015*; *Mao et al., 2000*; *Nicholas et al., 1989*; *Plewa, Potvin & Dickey, 2016*; *Raschner et al., 2010*), pronation/supination ($n = 6$; *Axelsson et al., 2018*; *Harbin, Leyh & Harbin, 2020*; *Kramer et al., 1993*; *Matsuoka et al., 2006*; *Nilsson et al., 2019*; *Puharic & Bohannon, 1993*), and ulnar/radial deviation ($n = 3$; *La Delfa et al., 2015*; *Miller, Nair & Baratz, 2005*; *Plewa, Potvin & Dickey, 2016*). Almost all articles evaluated movement direction pairs, with few articles looking at single movement directions: extension (*Andrews, Thomas & Bohannon, 1996*; *Cornwall, 1994*; *Richards, Gordon & Beaton, 1993*), flexion (*Harbin, Leyh & Harbin, 2020*; *Hill et al., 2018*; *Lorbergs et al., 2011*), and ulnar deviation (*Crawford, Wanibe &*

*Nayak, 2002*). Three included articles examined solely isokinetic movements (*Forthomme et al., 2002*; *Mao et al., 2000*; *Nicholas et al., 1989*), 18 examined isometric movements (*Andrews, Thomas & Bohannon, 1996*; *Axelsson et al., 2018*; *Cornu, Maietti & Ledoux, 2003*; *Cornwall, 1994*; *Crawford, Wanibe & Nayak, 2002*; *Hallbeck, 1994*; *Harbin, Leyh & Harbin, 2020*; *Holzbaur et al., 2007*; *Karahan et al., 2017*; *La Delfa et al., 2015*; *Lorbergs et al., 2011*; *Matsuoka et al., 2006*; *Miller, Nair & Baratz, 2005*; *Nilsson et al., 2019*; *Plewa, Potvin & Dickey, 2016*; *Puharic & Bohannon, 1993*; *Raschner et al., 2010*; *Richards, Gordon & Beaton, 1993*), and four articles assessed a combination of both movement types (*Danneskiold-Samsøe et al., 2009*; *Harbo, Brincks & Andersen, 2012*; *Hill et al., 2018*; *Kramer et al., 1993*).

## Measurement methodology

Various measurement tools were used to assess wrist strength. A handheld dynamometer was the most popular measurement tool ($n = 15$; *Andrews, Thomas & Bohannon, 1996*; *Axelsson et al., 2018*; *Crawford, Wanibe & Nayak, 2002*; *Danneskiold-Samsøe et al., 2009*; *Forthomme et al., 2002*; *Harbin, Leyh & Harbin, 2020*; *Harbo, Brincks & Andersen, 2012*; *Hill et al., 2018*; *Holzbaur et al., 2007*; *Karahan et al., 2017*; *Kramer et al., 1993*; *Lorbergs et al., 2011*; *Mao et al., 2000*; *Nicholas et al., 1989*; *Puharic & Bohannon, 1993*), with force transducers (*Hallbeck, 1994*; *Plewa, Potvin & Dickey, 2016*; *Raschner et al., 2010*), and load cells (*Cornwall, 1994*; *La Delfa et al., 2015*; *Miller, Nair & Baratz, 2005*) also commonly used. Less common measurement tools included an ergometer (*Cornu, Maietti & Ledoux, 2003*), a myometer (*Richards, Gordon & Beaton, 1993*), a force cell (*Matsuoka et al., 2006*), and a torque meter with a strain gauge (*Nilsson et al., 2019*). Almost all articles measured force in Newtons or torque as Newton-meters; only two articles used kg as dependent measure units (*Harbin, Leyh & Harbin, 2020*; *Richards, Gordon & Beaton, 1993*).

## Strength measurements

Wide ranges of strength outcomes were observed but collectively demonstrated decreased female strength relative to males; when normalized to male strength, variability existed between movement directions. These have been separated by movement planes below.

Sex differences in wrist flexion/extension strength were documented to have wide variability across articles but were centralized around females having ~60–65% of male strength (Tables 3 and 4). Extracted female strengths ranged from 38–135% and 42–144% of male strengths in flexion and extension, respectively (*Andrews, Thomas & Bohannon, 1996*; *Cornu, Maietti & Ledoux, 2003*; *Danneskiold-Samsøe et al., 2009*; *Forthomme et al., 2002*; *Hallbeck, 1994*; *Harbo, Brincks & Andersen, 2012*; *Hill et al., 2018*; *Holzbaur et al., 2007*; *Karahan et al., 2017*; *La Delfa et al., 2015*; *Lorbergs et al., 2011*; *Mao et al., 2000*; *Nicholas et al., 1989*; *Plewa, Potvin & Dickey, 2016*; *Raschner et al., 2010*; *Richards, Gordon & Beaton, 1993*) (Fig. 2). The greatest percentage disparity in strength occurred in *La Delfa et al. (2015)*, with females generating isometric torques of 4.62 Nm in flexion and 3.71 Nm in extension, compared to 12.28 Nm and 8.82 Nm respectively for males (*La Delfa et al., 2015*). Instances of female flexion strength exceeding male strength were observed in isokinetic flexion trials in a single article, where peak torques represented 133% and 135%

Peer J

**Table 3 Included articles with flexion outcome measures.** Included articles within this table contained outcome measures for flexion, and includes author information, participant information, measurement methods, type of contraction, outcome measures, and then calculated or included effect sizes.

| Article | Participants (males; females) | Measurement method | Type of contraction | Isokinetic speeds (If applicable) | Outcome values | Effect size (Cohen's d) |
|---|---|---|---|---|---|---|
| Cornu, Maietti & Ledoux (2003) | 5 males (29.5 ± 4.4yrs) <br> 3 females (24.0 ± 2.4yrs) | Ergometer | Isometric (Nm) | | Males: 18.34 ± 0.61 <br> Females: 9.58 ± 1.61 | $d = 14.36$ |
| Danneskiold-Samsøe et al. (2009) | 42 males <br> 77 females | Dynamometer | Isokinetic (Nm) <br> Isometric (N) | 30°/s <br> 60°/s <br> 90°/s | Males Isokinetic: <br> 20–29: <br> 30°/s: 21.1 ± 4.6 <br> 60°/s: 19.3 ± 4.7 <br> 90°/s: 18.4 ± 3.8 <br> 30–39: <br> 30°/s: 21.9 ± 4.5 <br> 60°/s: 20.6 ± 4.4 <br> 90°/s: 18.9 ± 4.9 <br> 40–49: <br> 30°/s: 21.5 ± 4.2 <br> 60°/s: 21.2 ± 4.9 <br> 90°/s: 20.3 ± 4.5 <br> 50–59: <br> 30°/s: 22.2 ± 5.1 <br> 60°/s: 21.2 ± 4.7 <br> 90°/s: 19.8 ± 4.0 <br> Females Isokinetic: <br> 20–29: <br> 30°/s: 12.4 ± 3.1 <br> 60°/s: 11.5 ± 2.7 <br> 90°/s: 10.7 ± 2.5 <br> 30–39: <br> 30°/s: 12.3 ± 3.0 <br> 60°/s: 11.7 ± 2.8 <br> 90°/s: 11.1 ± 2.7 <br> 40–49: <br> 30°/s: 12.8 ± 3.3 <br> 60°/s: 12.4 ± 2.7 <br> 90°/s: 11.5 ± 2.3 <br> 50–59: <br> 30°/s: 12.5 ± 2.3 <br> 60°/s: 12.0 ± 2.2 <br> 90°/s: 11.7 ± 1.9 <br> Males Isometric: <br> 20–29: 23.9 ± 7.4 <br> 30–39: 23.6 ± 2.8 <br> 40–49: 25.4 ± 5.0 <br> 50–59: 25.9 ± 5.6 <br> Females Isometric: <br> 20–29: 14.3 ± 3.8 <br> 30–39: 14.1 ± 2.6 <br> 40–49: 15.0 ± 3.9 <br> 50–59: 14.6 ± 2.9 | Isokinetic: <br> 20–29: <br> 30°/s: $d = 1.89$ <br> 60°/s: $d = 1.66$ <br> 90°/s: $d = 2.03$ <br> 30–39: <br> 30°/s: $d = 2.1$ <br> 60°/s: $d = 2.02$ <br> 90°/s: $d = 1.59$ <br> 40–49: <br> 30°/s: $d = 2.07$ <br> 60°/s: $d = 1.80$ <br> 90°/s: $d = 1.96$ <br> 50–59: <br> 30°/s: $d = 1.90$ <br> 60°/s: $d = 1.96$ <br> 90°/s: $d = 2.03$ <br> Isometric: <br> 20–29: $d = 1.30$ <br> 30–39: $d = 3.39$ <br> 40–49: $d = 2.08$ <br> 50–59: $d = 2.02$ |

*(continued on next page)*

Napper et al. (2023), PeerJ, DOI 10.7717/peerj.16557

**Table 3** (*continued*)

| Article | Participants (males; females) | Measurement method | Type of contraction | Isokinetic speeds (If applicable) | Outcome values | Effect size (Cohen's *d*) |
|---|---|---|---|---|---|---|
| *Forthomme et al. (2002)* | 20 males (23 ± 3yrs) 20 females (23 ± 3yrs) | Dynamometer | Isokinetic (Nm; Nm/kg) | 90°/s 30°/s 60°/s | Males Peak: 90°/s: 14.6 ± 3.37 30°/s: 15.7 ± 4.3 60°/s: 19.3 ± 5.7 Females Peak: 90°/s: 10.6 ± 2.0 30°/s: 11.3 ± 2.3 60°/s: 17 ± 4.6 Males Normalized: 90°/s: 0.26 ± 0.06 30°/s: 0.28 ± 0.08 60°/s: 0.35 ± 0.11 Females Normalized: 90°/s: 0.14 ± 0.02 30°/s: 0.15 ± 0.03 60°/s: 0.23 ± 0.08 | Peak: 90°/s: $d = 2.69$ 30°/s: $d = 3.71$ 60°/s: $d = 2.88$ Normalized: 90°/s: $d = 2.25$ 30°/s: $d = 1.40$ 60°/s: $d = 1.50$ |
| *Hallbeck (1994)* | 30 males 30 females | Force transducer | Isometric (N) | | Males: 90° flexion: 104.24 75° flexion: 101.05 60° flexion: 76.99 45° flexion: 81.60 30° flexion: 81.06 15° flexion: 76.36 15° extension: 74.78 30° extension: 73.21 45° extension: 65.74 60° extension: 57.54 75° extension: 47.54 90° extension: 60.55 Average over all postures: 73.15 Females: 90° flexion: 75.23 75° flexion: 79.89 60° flexion: 58.05 45° flexion: 59.76 30° flexion: 63.16 15° flexion: 57.15 15° extension: 58.25 30° extension: 54.01 45° extension: 48.28 60° extension: 43.06 75° extension: 39.11 90° extension: 47.05 Average over all postures: 55.42 | No SD reported |
**Table 3** (*continued*)

| Article | Participants (males; females) | Measurement method | Type of contraction | Isokinetic speeds (If applicable) | Outcome values | Effect size (Cohen's $d$) |
|---|---|---|---|---|---|---|
| *Harbo, Brincks & Andersen (2012)* | 93 males<br>85 females | Dynamometer | Isokinetic (Nm)<br>Isometric (Nm) | 90°/s | Males<br>Isokinetic: 22.1 ± 5.51<br>Isometric: 24.7 ± 7.45<br>Females:<br>Isokinetic: 13.1 ± 2.81<br>Isometric: 14.4 ± 3.90 | Isokinetic: $d = 1.63$<br>Isometric: $d = 1.38$ |
| *Hill et al. (2018)* | 12 males (22.8 ± 2.3yrs)<br>12 females (23 ± 3.2yrs) | Dynamometer | Isokinetic (Nm) | 60°/s | Males: 81.9 ± 16.7<br>Females: 44.1 ± 5.4 | $d = 2.26$ |
| *Holzbaur et al. (2007)* | 5 males (29.2 ± 4.4yrs)<br>5 females (28 ± 5.1yrs) | Biodex | Isometric (Nm) | | Males: 25.6 ± 7<br>Females: 10.7 ± 2.7 | $d = 2.13$ |
| *Karahan et al. (2017)* | 30 males (20.17 ± 1.0yrs)<br>34 females (19.71 ± 0.8yrs) | Dynamometer | Isometric (N) | ' | Males:<br>Right: 36.21 ± 12.16<br>Left: 31.82 ± 9.74<br>Females:<br>Right: 21.63 ± 8.52<br>Left: 18.24 ± 7.16 | Right: $d = 1.12$<br>Left: $d = 1.39$ |
| *La Delfa et al. (2015)* | 12 males (23.8 ± 8.1yrs)<br>12 females (21.8 ± 2.2yrs) | Load cell | Isometric (Nm) | | Males:<br>Pronation: 12.28 ± 2.49<br>Supination: 10.10 ± 1.84<br>Females:<br>Pronation: 4.62 ± 1.57<br>Supination: 4.14 ± 1.19 | Pronation: $d = 3.08$<br>Supination: $d = 3.24$ |
| *Lorbergs et al. (2011)* | 23 males (50.8 ± 1.2yrs)<br>17 females (47.8 ± 2.1yrs) | Dynamometer | Isometric (Nm) | | Males: 21.6 ± 8.8<br>Females: 8.8 ± 3.4 | $d = 1.45$ |

Peer

**Table 3** (*continued*)

| Article | Participants (males; females) | Measurement method | Type of contraction | Isokinetic speeds (If applicable) | Outcome values | Effect size (Cohen's *d*) |
|---|---|---|---|---|---|---|
| *Mao et al. (2000)* | 35 males (20 ± 1.2yrs) 34 females (20.1 ± 1.3yrs) | Dynamometer | Isokinetic (Nm; Nm/kg) | 30°/s 60°/s 120°/s 180°/s 240°/s | Males: Peak torque: 30°/s: 13.06 ± 3.0 60°/s: 10.86 ± 2.9 120°/s: 10.14 ± 2.46 180°/s: 8.74 ± 2.47 240°/s: 7.63 ± 2.2 Relative peak torque: 30°/s: 7.27 ± 1.87 60°/s: 6.17 ± 1.84 120°/s: 5.40 ± 1.33 180°/s: 5.09 ± 1.40 240°/s: 4.51 ± 1.29 Females: Peak torque: 30°/s: 7.27 ± 1.87 60°/s: 6.17 ± 1.84 120°/s: 5.40 ± 1.33 180°/s: 5.09 ± 1.40 240°/s: 4.51 ± 1.29 Relative peak torque: 30°/s: 0.13 ± 0.03 60°/s: 0.11 ± 0.03 120°/s: 0.10 ± 0.02 180°/s: 0.09 ± 0.02 240°/s: 0.08 ± 0.02 | Peak torque: 30°/s: *d* = 1.93 60°/s: *d* = 1.62 120°/s: *d* = 1.93 180°/s: *d* = 1.48 240°/s: *d* = 1.42 Relative peak torque: 30°/s: *d* = 1.60 60°/s: *d* = 1.20 120°/s: *d* = 1.50 180°/s: *d* = 1.25 240°/s: *d* = 1.0 |
| *Nicholas et al. (1989)* | 9 males 20 females | Dynamometer | Isokinetic (Nm) | 60°/s 120°/s | Males: Peak torque: 60°/s: 4.3 ± 3.0 120°/s: 2.6 ± 2.0 Relative peak torque: 60°/s: 7.0 ± 10.4 120°/s: 2.2 ± 1.6 Females: Peak torque: 60°/s: 5.7 ± 5.0 120°/s: 3.5 ± 3.0 Relative peak torque: 60°/s: 4.2 ± 3.9 120°/s: 2.4 ± 2.4 | Peak torque: 60°/s: *d* = 0.47 120°/s: *d* = 0.45 Relative peak torque: 60°/s: *d* = 0.27 120°/s: *d* = 0.13 |
| *Plewa, Potvin & Dickey (2016)* | 14 males (24.3 ± 2.6yrs) 14 females (24.6 ± 2.4yrs) | Force/torque transducer | Isometric (Nm) | | Males: 4.13 ± 1.70 Females: 2.92 ± 1.53 | *d* = 0.71 |
| *Raschner et al. (2010)* | 19 males 17 females | Force transducer | Isometric (N) | | Males: 373 ± 72 Females: 243 ± 52 | *d* = 1.81 |

Napper et al. (2023), *PeerJ*, DOI 10.7717/peerj.16557

**Table 4  Included articles with extension outcome measures.** Included articles within this table contained outcome measures for extension, and includes author information, participant information, measurement methods, type of contraction, outcome measures, and then calculated or included effect sizes.

| Article | Participants (males; females) | Measurement method | Type of contraction | Isokinetic speeds (if applicable) | Outcome values | Effect size (Cohen's $d$) |
|---|---|---|---|---|---|---|
| *Andrews, Thomas & Bohannon (1996)* | 25 males (54 ± 3.4yrs) 25 females (54 ± 2.8yrs) | Dynamometer | Isometric (N) | | Males: 149.1 ± 31.1 Females: 90.8 ± 21.9 | $d = 1.86$ |
| *Cornu, Maietti & Ledoux (2003)* | 5 males (29.5 ± 4.4yrs) 3 females (24 ± 2.4yrs) | Ergometer | Isometric (Nm) | | Males: 12.84 ± 3.42 Females: 5.93 ± 1.23 | $d = 2.02$ |
| *Danneskiold-Samsøe et al. (2009)* | 42 males 77 females | Dynamometer | Isokinetic (Nm) Isometric (N) | 30°/s 60°/s 90°/s | Males Isokinetic: 20–29: 30°/s: 11.0 ± 2.5 60°/s: 10.4 ± 3.3 90°/s: 9.4 ± 1.6 30–39: 30°/s: 11.3 ± 1.3 60°/s: 11.0 ± 1.2 90°/s: 10.1 ± 1.6 40–49: 30°/s: 12.4 ± 2.4 60°/s: 11.9 ± 2.7 90°/s: 11.5 ± 2.7 50–59: 30°/s: 11.8 ± 3.7 60°/s: 11.0 ± 3.2 90°/s: 11.4 ± 3.9 Females Isokinetic: 20–29: 30°/s: 6.4 ± 1.4 60°/s: 5.9 ± 1.8 90°/s: 5.7 ± 1.4 30–39: 30°/s: 6.4 ± 1.7 60°/s: 5.9 ± 1.8 90°/s: 5.9 ± 1.8 40–49: 30°/s: 7.1 ± 1.9 60°/s: 6.7 ± 1.9 90°/s: 6.3 ± 2.0 50–59: 30°/s: 5.9 ± 1.4 60°/s: 5.8 ± 1.3 90°/s: 5.6 ± 1.2 Males Isometric: 20–29: 13.1 ± 3.0 30–39: 13.9 ± 2.6 40–49: 14.4 ± 2.8 50–59: 12.7 ± 2.5 Females Isometric: 20–29: 6.9 ± 2.0 30–39: 7.3 ± 2.3 40–49: 7.5 ± 2.2 50–59: 5.8 ± 1.7 | Isokinetic: 20–29: 30°/s: $d = 1.84$ 60°/s: $d = 2.05$ 90°/s: $d = 2.31$ 30–39: 30°/s: $d = 3.77$ 60°/s: $d = 4.25$ 90°/s: $d = 2.63$ 40–49: 30°/s: $d = 2.21$ 60°/s: $d = 1.93$ 90°/s: $d = 1.93$ 50–59: 30°/s: $d = 1.59$ 60°/s: $d = 1.63$ 90°/s: $d = 1.49$ Isometric: 20–29: $d = 2.07$ 30–39: $d = 2.54$ 40–49: $d = 2.46$ 50–59: $d = 2.76$ |

| Article | Participants (males; females) | Measurement method | Type of contraction | Isokinetic speeds (if applicable) | Outcome values | Effect size (Cohen's *d*) |
|---|---|---|---|---|---|---|
| *Forthomme et al. (2002)* | 20 males (23 ± 3yrs) 20 females (23 ± 3yrs) | Dynamometer | Isokinetic (Nm; Nm/kg) | 90°/s 30°/s 60°/s | Males peak: 90°/s: 10.6 ± 2.0 30°/s: 11.3 ± 2.3 60°/s: 17 ± 4.6 Females peak: 90°/s: 6.9 ± 1.8 30°/s: 7.6 ± 1.6 60°/s: 15.4 ± 3.6 Males normalized: 90°/s: 0.14 ± 0.02 30°/s: 0.15 ± 0.03 60°/s: 0.23 ± 0.08 Females normalized: 90°/s: 0.12 ± 0.03 30°/s: 0.12 ± 0.02 60°/s: 0.28 ± 0.06 | Peak torque: 90°/s: *d* = 1.85 30°/s: *d* = 1.61 60°/s: *d* = 0.35 Normalized torque: 90°/s: *d* = 1.0 30°/s: *d* = 1.0 60°/s: *d* = 0.63 |
| *Hallbeck (1994)* | 30 males 30 females | Force transducer | Isometric (N) | | Males: 90° flexion: 68.54 75° flexion: 58.86 60° flexion: 51.79 45° flexion: 64.44 30° flexion: 66.77 15° flexion: 66.44 15° extension: 68.35 30° extension: 68.36 45° extension: 60.35 60° extension: 52.24 75° extension: 54.29 90° extension: 76.87 Average over all postures: 62.29 Females: 90° flexion: 45.49 75° flexion: 46.12 60° flexion: 40.72 45° flexion: 46.49 30° flexion: 47.68 15° flexion: 45.56 15° extension: 46.90 30° extension: 47.01 45° extension: 42.31 60° extension: 37.46 75° extension: 40.46 90° extension: 59.03 Average over all postures: 45.05 | No SD reported |
| *Harbo, Brincks & Andersen (2012)* | 93 males 85 females | Dynamometer | Isokinetic (Nm) | 90°/s | Males: 11.4 ± 3.12 Females: 6.01 ± 1.68 | *d* = 1.73 |
| *Holzbaur et al. (2007)* | 5 males (29.2 ± 4.4yrs) 5 females (28 ± 5.1yrs) | Biodex | Isometric (Nm) | | Males: 14.0 ± 3.4 Females: 6.4 ± 0.9 | *d* = 2.24 |
| *Karahan et al. (2017)* | 30 males (20.17 ± 0.1yrs) 34 females (19.71 ± 0.8yrs) | Dynamometer | Isometric (N) | | Males: Right: 31.65 ± 9.15 Left: 28.54 ± 8.52 Females: Right: 17.13 ± 7.07 Left: 14.68 ± 6.57 | Right: *d* = 1.59 Left: *d* = 1.63 |
**Table 4** (*continued*)

| Article | Participants (males; females) | Measurement method | Type of contraction | Isokinetic speeds (if applicable) | Outcome values | Effect size (Cohen's *d*) |
|---|---|---|---|---|---|---|
| *La Delfa et al. (2015)* | 12 males (23.8 ± 8.1yrs) 12 females (21.8 ± 2.2yrs) | Load cell | Isometric (Nm) | | Males: Pronation: 8.82 ± 2.07 Supination: 6.52 ± 1.56 Females: Pronation: 3.71 ± 1.25 Supination: 2.71 ± 0.89 | Pronation: *d* = 2.47 Supination: *d* = 2.44 |
| *Mao et al. (2000)* | 35 males (20.1 ± 1.2yrs) 34 females (20.1 ± 1.3yrs) | Dynamometer | Isokinetic (Nm; Nm/kg) | 30°/s 60°/s 120°/s 180°/s 240°/s | Males: Peak torque: 30°/s: 9.94 ± 2.30 60°/s: 8.80 ± 2.19 120°/s: 8.17 ± 2.18 180°/s: 7.29 ± 2.16 240°/s: 6.91 ± 2.03 Relative peak torque: 30°/s: 0.16 ± 0.04 60°/s: 0.14 ± 0.04 120°/s: 0.13 ± 0.04 180°/s: 0.11 ± 0.04 240°/s: 0.10 ± 0.03 Females: Peak torque: 30°/s: 6.62 ± 1.65 60°/s: 6.20 ± 1.57 120°/s: 5.11 ± 1.51 180°/s: 4.54 ± 1.48 240°/s: 4.26 ± 1.17 Relative peak torque: 30°/s: 0.23 ± 0.60 60°/s: 0.11 ± 0.03 120°/s: 0.09 ± 0.03 180°/s: 0.08 ± 0.03 240°/s: 0.07 ± 0.02 | Peak torque: 30°/s: *d* = 1.44 60°/s: *d* = 1.19 120°/s: *d* = 1.40 180°/s: *d* = 1.27 240°/s: *d* = 1.31 Relative peak torque: 30°/s: *d* = 1.75 60°/s: *d* = 0.75 120°/s: *d* = 1.0 180°/s: *d* = 0.75 240°/s: *d* = 1.0 |
| *Nicholas et al. (1989)* | 9 males 20 females | Dynamometer | Isokinetic (Nm) | 60°/s 120°/s | Males: Peak torque: 60°/s: 5.7 ± 2.0 120°/s: 4.6 ± 1.0 Relative peak torque: 60°/s: 3.4 ± 1.0 120°/s: 3.4 ± 1.0 Females: Peak torque: 60°/s: 5.8 ± 3.0 120°/s: 2.6 ± 1.0 Relative peak torque: 60°/s: 4.3 ± 2.3 120°/s: 2.0 ± 1.20 | Peak torque: 60°/s: *d* = 0.05 120°/s: *d* = 1.0 Relative peak torque: 60°/s: *d* = 0.90 120°/s: *d* = 1.40 |
| *Plewa, Potvin & Dickey (2016)* | 14 males (24.3 ± 2.6yrs) 14 females (24.6 ± 2.4yrs) | Force/torque transducer | Isometric (Nm) | | Males: 3.03 ± 1.39 Females: 2.16 ± 1.09 | *d* = 0.63 |
| *Raschner et al. (2010)* | 19 males 17 females | Force transducers | Isometric (N) | | Males: 159 ± 25 Females: 106 ± 22 | *d* = 2.12 |

Napper et al. (2023), *PeerJ*, DOI 10.7717/peerj.16557

| Article | Participants (males; females) | Measurement method | Type of contraction | Isokinetic speeds (if applicable) | Outcome values | Effect size (Cohen's *d*) |
|---------|-------------------------------|---------------------|---------------------|-----------------------------------|----------------|---------------------------|
| *Richards, Gordon & Beaton (1993)* | 132 males<br>167 females | Myometer | Isometric (kg) | | Males:<br>20–29: 16.0<br>30–39: 15.4<br>40–49: 15.0<br>50–59: 14.0<br>Females:<br>20–29: 8.9<br>30–39: 8.1<br>40–49: 7.4<br>50–59: 7.4 | No SD reported |

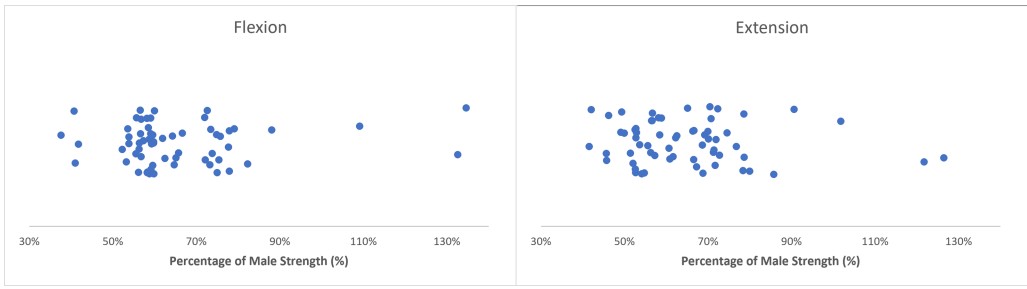

**Figure 2  Normative flexion/extension strengths.** Sex differences in flexion (left) and extension (right) strength with female strength metrics ranging from 38–135% and 42–144% of male strength, respectively. Articles in this review indicate that female flexion/extension strength centralized at ∼60–65% of male strength.

of male torques when normalized to body weight (5.7 Nm *vs* 4.3 Nm at 60°/s, 3.5 Nm *vs* 2.6 Nm at 120°/s for females and males, respectively) (*Nicholas et al., 1989*). This article is the only instance of female flexion strength exceeding male strength; the next closest article had females generating 88% of male eccentric isokinetic flexion strength at 60°/s (*Forthomme et al., 2002*). Female extension strengths occupied similar ranges of male strength as flexion outcome measures but appeared more dispersed within the 40–80% strength ranges. Two articles had extension outcome measures where female strengths exceeded males. In both instances, females exceeded male strengths which occurred in isokinetic exertions at lower speeds; once in concentric extension at 30°/s when normalized to body weight (0.23 Nm/kg and 0.16 Nm/kg for females and males, respectively) (*Mao et al., 2000*) and at 60°/s in body-weight normalized (4.3 Nm/kg and 3.4 Nm/kg for females and males respectively) and non-normalized outcome measures (5.8 Nm and 5.7 Nm for females and males, respectively) (*Nicholas et al., 1989*).

Moderately variable differences in wrist ulnar/radial deviation were presented, centralized with females producing ∼60–70% of male strength (Tables 5 and 6). Extracted female strengths ranged from 48–80% and 51–80% of male strengths in radial and ulnar deviations, respectively (*Crawford, Wanibe & Nayak, 2002*; *La Delfa et al., 2015*; *Miller, Nair & Baratz, 2005*; *Plewa, Potvin & Dickey, 2016*) (Fig. 3). The greatest ulnar/radial strength differences were observed during isometric contractions when paired with supination/pronation (*La Delfa et al., 2015*). Females produced their greatest ulnar strength when paired with a supinated forearm, generating 67% of male strength (*La Delfa et al., 2015*) (Table 5). Alternatively, females generated their greatest radial strength when paired with a pronated forearm, producing 66% of male strength (*La Delfa et al., 2015*) (Table 6). One article demonstrated increased female ability to generate both ulnar and radial deviation strength, producing >80% of male radial isometric strength, but this was the only article that demonstrated these values (*Miller, Nair & Baratz, 2005*).

Sex differences in pronation/supination strength were documented to have less variability across articles when compared to other movement patterns, with females generating ∼55–60% of male strength (Tables 7 and 8). Extracted female strengths ranged from 46–98%

Napper et al. (2023), *PeerJ*, DOI 10.7717/peerj.16557

**Table 5 Included articles with ulnar deviation outcome measures.** Included articles within this table contained outcome measures for ulnar deviation, and includes author information, participant information, measurement methods, type of contraction, outcome measures, and then calculated or included effect sizes.

| Article | Participants (males; females) | Measurement method | Type of contraction | Isokinetic speeds (if applicable) | Outcome values | Effect size (Cohen's $d$) |
|---|---|---|---|---|---|---|
| *Crawford, Wanibe & Nayak (2002)* | 10 males (26 ± 4.2yrs) 10 females (25.9 ± 4.8yrs) | Dynamometer | Isometric (Nm) | | Males: A: 1.76 ± 0.31 B: 2.75 ± 0.43 C: 2.98 ± 1.04 D: 4.22 ± 0.85 E: 4.95 ± 0.88 F: 5.49 ± 1.09 G: 6.49 ± 1.36 H: 6.07 ± 1.08 I: 6.28 ± 1.10 J: 5.21 ± 1.22 K: 5.30 ± 0.80 L: 6.56 ± 1.42 Females: A: 1.19 ± 0.40 B: 1.84 ± 0.54 C: 2.19 ± 0.44 D: 3.04 ± 0.81 E: 3.09 ± 0.93 F: 3.27 ± 1.13 G: 3.84 ± 1.45 H: 3.83 ± 1.33 I: 3.96 ± 1.35 J: 3.44 ± 1.08 K: 4.01 ± 1.29 L: 3.87 ± 1.36 | A: $d = 1.84$ B: $d = 2.12$ C: $d = 0.76$ D: $d = 1.39$ E: $d = 2.11$ F: $d = 2.02$ G: $d = 1.91$ H: $d = 2.07$ I: $d = 2.11$ J: $d = 1.45$ K: $d = 1.61$ L: $d = 1.89$ |
| *La Delfa et al. (2015)* | 12 males (23.8 ± 8.1yrs) 12 females (21.8 ± 2.2yrs) | Tri-axial load cell | Isometric (Nm) | | Males: Pronation: 5.86 ± 0.77 Supination: 7.77 ± 2.02 Females: Pronation: 3.95 ± 1.57 Supination: 3.97 ± 1.64 | Pronation: $d = 2.48$ Supination: $d = 1.88$ |

Napper et al. (2023), *PeerJ*, DOI 10.7717/peerj.16557

**Table 5** (*continued*)

| Article | Participants (males; females) | Measurement method | Type of contraction | Isokinetic speeds (if applicable) | Outcome values | Effect size (Cohen's *d*) |
|---|---|---|---|---|---|---|
| *Miller, Nair & Baratz (2005)* | 18 males (39 ± 13yrs) 46 females (41 ± 12yrs) | Load cell | Isometric (Nm) | | Males: 2.5 cm disk: 0.63 ± 0.12 5 cm disk: 2.16 ± 0.63 7.5 cm disk: 3.37 ± 0.86 10 cm disk: 4.26 ± 1.17 12.5 cm disk: 4.69 ± 1.32 Females: 2.5 cm disk: 0.50 ± 0.14 5 cm disk: 1.44 ± 0.41 7.5 cm disk: 2.20 ± 0.63 10 cm disk: 2.79 ± 0.82 12.5 cm disk: 2.75 ± 0.90 | 2.5 cm disk: *d* = 1.08 5 cm disk: *d* = 1.14 7.5 cm disk: *d* = 1.36 10 cm disk: *d* = 1.26 12.5 cm disk: *d* = 1.47 |
| *Plewa, Potvin & Dickey (2016)* | 14 males (24.3 ± 2.6yrs) 14 females (24.6 ± 2.4yrs) | Force/torque transducer | Isometric (Nm) | | Males: 7.52 ± 2.67 Females: 4.52 ± 1.95 | *d* = 1.12 |

Napper et al. (2023), PeerJ, DOI 10.7717/peerj.16557

Peerj

**Table 6  Included articles with radial deviation outcome measures.** Included articles within this table contained outcome measures for radial deviation, and includes author information, participant information, measurement methods, type of contraction, outcome measures, and then calculated or included effect sizes.

| Article | Participants (males; females) | Measurement method | Type of contraction | Isokinetic speeds (if applicable) | Outcome values | Effect size (Cohen's $d$) |
|---|---|---|---|---|---|---|
| La Delfa et al. (2015) | 12 males (23.8 ± 8.1yrs) 12 females (21.8 ± 2.2yrs) | Tri-axial load cell | Isometric (Nm) | | Males: Pronation: 9.36 ± 1.52 Supination: 5.41 ± 0.77 Females: Pronation: 4.51 ± 1.76 Supination: 3.56 ± 1.37 | Pronation: $d = 3.21$ Supination: $d = 2.40$ |
| Miller, Nair & Baratz (2005) | 18 males (39 ± 13yrs) 46 females (41 ± 12yrs) | Load cell | Isometric (Nm) | | Males: 2.5 cm disk: 0.61 ± 0.15 5 cm disk: 2.42 ± 0.81 7.5 cm disk: 3.46 ± 1.24 10 cm disk: 4.29 ± 1.53 12.5 cm disk: 4.76 ± 1.32 Females: 2.5 cm disk: 0.49 ± 0.14 5 cm disk: 1.60 ± 0.41 7.5 cm disk: 2.17 ± 0.62 10 cm disk: 2.62 ± 0.68 12.5 cm disk: 2.62 ± 0.81 | 2.5 cm disk: $d = 0.80$ 5 cm disk: $d = 1.01$ 7.5 cm disk: $d = 1.04$ 10 cm disk: $d = 1.09$ 12.5 cm disk: $d = 1.62$ |
| Plewa, Potvin & Dickey (2016) | 14 males (24.3 ± 2.6yrs) 14 females (24.6 ± 2.4yrs) | Force/torque transducer | Isometric (Nm) | | Males: 5.81 ± 2.52 Females: 3.48 ± 2.04 | $d = 0.92$ |

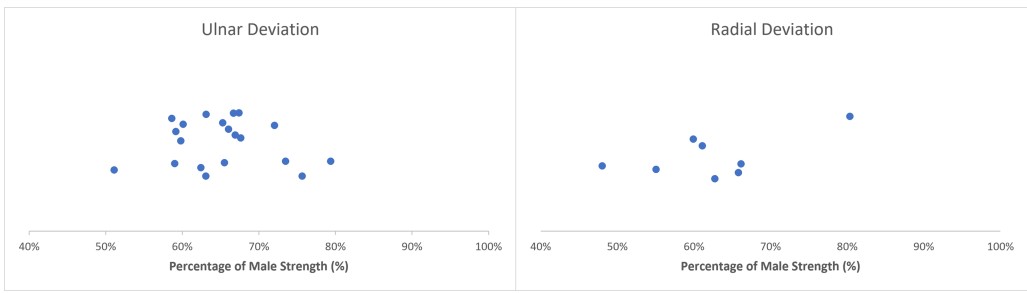

**Figure 3   Normative ulnar/radial deviation strengths.** Sex differences in ulnar (left) and radial (right) deviation strength with female strength metrics ranging from 51–80% and 48–80% of male strength, respectively. Articles in this review indicate that female ulnar/radial deviation strength centralized at 60–70% of male strength.

and 45–95% of male strengths in pronation and supination, respectively (*Axelsson et al., 2018*; *Harbin, Leyh & Harbin, 2020*; *Kramer et al., 1993*; *Matsuoka et al., 2006*; *Nilsson et al., 2019*; *Puharic & Bohannon, 1993*) (Fig. 4). The greatest percentage disparity in pronation strength occurred when females generated isometric torques of 3.33 Nm when in complete pronation and 5.10 Nm while pronating in a supinated posture, where males produced 7.24 Nm and 10.12 Nm, respectively (*Matsuoka et al., 2006*). Substantial supination differences were observed during neutral isometric contractions, where females 18–35 yrs produced torques of 4.1 Nm and 3.8 Nm at 36–65 yrs, where males generated 6.1 Nm and 8.4 Nm, respectively (*Nilsson et al., 2019*). Minimal sex differences were observed in non-dominant hand contractions, where females were able to generate 94 N of isometric force compared to 96 N in males (*Axelsson et al., 2018*); this was the only article that displayed such similar values. Similar values between sexes were only observed once in isokinetic trials at 60°/s and 120°/s, where females produced 9.7 Nm and 9.3 Nm, where males generated 10.2 Nm and 9.8 Nm, respectively (*Kramer et al., 1993*).

## Risk of bias assessment

Risk of bias assessments of each article conducted using the ROBINS-I identified one article with severe risk of bias due to missing data (*Nicholas et al., 1989*); 12 articles had moderate risks of bias across multiple domains, including bias due to confounding (*Axelsson et al., 2018*), selection of participants (*Danneskiold-Samsøe et al., 2009*; *Harbin, Leyh & Harbin, 2020*; *Harbo, Brincks & Andersen, 2012*; *Holzbaur et al., 2007*; *Karahan et al., 2017*; *La Delfa et al., 2015*; *Mao et al., 2000*; *Nicholas et al., 1989*; *Raschner et al., 2010*), deviations from intended intervention (*Richards, Gordon & Beaton, 1993*), missing data (*Harbo, Brincks & Andersen, 2012*; *Hill et al., 2018*; *La Delfa et al., 2015*; *Matsuoka et al., 2006*), measurement outcomes (*Harbo, Brincks & Andersen, 2012*; *Hill et al., 2018*; *Nicholas et al., 1989*), and the selection of the reported results (*Hill et al., 2018*). Risk of bias was also prevalent in inequal participant ratios (*Cornu, Maietti & Ledoux, 2003*; *Danneskiold-Samsøe et al., 2009*; *Harbin, Leyh & Harbin, 2020*; *Harbo, Brincks & Andersen, 2012*; *Lorbergs et al., 2011*; *Miller, Nair & Baratz, 2005*; *Nilsson et al., 2019*; *Raschner et al., 2010*; *Richards, Gordon & Beaton, 1993*).

**Table 7  Included articles with pronation outcome measures.** Included articles within this table contained outcome measures for pronation, and includes author information, participant information, measurement methods, type of contraction, outcome measures, and then calculated or included effect sizes.

| Article | Participants (males; females) | Measurement method | Type of contraction | Isokinetic speeds (if applicable) | Outcome values | Effect size (Cohen's $d$) |
|---|---|---|---|---|---|---|
| Axelsson et al. (2018) | 262 males (41 ± 18yrs) 237 females (47 ± 18yrs) | Dynamometer | Isometric (Nm, N) | | Males: Right torque: 7.9 ± 2.2 Left torque: 7.6 ± 7.8 Right lift: 157 ± 62 Left lift: 96 ± 41 Females: Right torque: 4.5 ± 1.2 Left torque: 4.3 ± 1.2 Right lift: 96 ± 41 Left lift: 94 ± 40 | Right torque: $d = 1.55$ Left torque: $d = 0.42$ Right lift: $d = 26.52$ Lift left: $d = 0.05$ |
| Harbin, Leyh & Harbin (2020) | 61,504 males 32,917 females | Dynamometer | Isometric (kg) | | Males: 20–29: R: 3.32; L:3.26 30–39: R: 3.52; L: 3.49 40–49: R: 3.53; L: 3.53 50–59: R: 3.42; L: 3.45 Females: 20–29: R: 1.87; L: 1.83 30–39: R: 2.03; L: 1.99 40–49: R: 2.06; L: 2.04 50–59: R: 1.95; L: 1.96 | No SD reported |
| Kramer et al. (1993) | 21 males (29.7 ± 6.9yrs) 22 females (28.4 ± 5.7yrs) | Dynamometer | Isokinetic (Nm) | 0°/s 60°/s 120°/s | Males: 0°/s: 12.6 ± 3.2 60°/s: 12.4 ± 2.5 120°/s: 11.7 ± 2.5 Females: 0°/s: 7.2 ± 1.9 60°/s: 7.1 ± 1.4 120°/s: 6.5 ± 1.5 | 0°/s: $d = 1.69$ 60°/s: $d = 2.12$ 120°/s: $d = 2.08$ |
| Matsuoka et al. (2006) | 23 males 27 females | Torque cell | Isometric (Nm) | | Males: Pronated: 7.24 ± 3.8 Supinated: 10.12 ± 3.27 Females: Pronated: 3.33 ± 1.48 Supinated: 5.10 ± 1.24 | Pronation: $d = 1.03$ Supination; $d = 1.54$ |

Napper et al. (2023), *PeerJ*, DOI 10.7717/peerj.16557

**Table 7** (*continued*)

| Article | Participants (males; females) | Measurement method | Type of contraction | Isokinetic speeds (if applicable) | Outcome values | Effect size (Cohen's *d*) |
|---|---|---|---|---|---|---|
| *Nilsson et al. (2019)* | 14 males (24.3 ± 2.6yrs) 14 females (24.6 ± 2.4yrs) | Force/torque transducer | Isometric (Nm) | | Males: 18–35: 5.7 ± 1.8 36–65: 7.0 ± 1.4 Females: 18–35: 3.8 ± 0.9 36–65: 3.7 ± 0.8 | 18–35: *d* = 1.06 36–65: *d* = 2.36 |
| *Puharic & Bohannon (1993)* | 12 males 12 females | Dynamometer | Isometric (N) | | Males: 39 ± 10.7 Females:19.3 ± 4.7 | *d* = 1.84 |

**Table 8 Included articles with supination outcome measures.** Included articles within this table contained outcome measures for supination, and includes author information, participant information, measurement methods, type of contraction, outcome measures, and then calculated or included effect sizes.

| Article | Participants (males; females) | Measurement method | Type of contraction | Isokinetic speeds (if applicable) | Outcome values | Effect size (Cohen's $d$) |
|---|---|---|---|---|---|---|
| Axelsson et al. (2018) | 262 males (41 ± 18yrs) 237 females (47 ± 18yrs) | Dynamometer | Isometric (Nm; N) | | Males: Right torque: 9.1 ± 2.3 Left torque: 8.9 ± 2.3 Right lift: 238 ± 89 Left lift: 230 ± 89 Females: Right torque: 5.4 ± 1.3 Left torque: 8.9 ± 2.3 Right lift: 137 ± 58 Left lift: 132 ± 55 | Right torque: $d = 1.61$ Left torque: $d = 1.61$ Right lift: $d = 1.13$ Left lift: $d = 1.10$ |
| Harbin, Leyh & Harbin (2020) | 61,504 males 32,917 females | Dynamometer | Isometric (kg) | | Males: 20–29: R: 3.14; L: 3.01 30–39: R: 3.20; L: 3.08 40–49: R: 3.05; L: 2.96 50–59: R:2.77; L: 2.69 Females: 20–29: R: 1.80; L: 1.70 30–39: R: 186; L: 1.76 40–49: R: 1.79; L: 1.72 50–59: R: 1.62; L: 1.55 | No SD reported |
| Kramer et al. (1993) | 21 males (29.7 ± 6.9yrs) 22 females (28.4 ± 5.7yrs) | Dynamometer | Isokinetic (Nm) | 0°/s 60°/s 120°/s | Males: 0°/s: 10.6 ± 2.5 60°/s: 10.2 ± 2.2 120°/s: 9.8 ± 2.1 Females: 0°/s: 5.8 ± 1.2 60°/s: 9.7 ± 2.2 120°/s: 9.3 ± 1.5 | 0°/s: $d = 1.92$ 60°/s: $d = 0.23$ 120°/s: $d = 0.24$ |
| Matsuoka et al. (2006) | 23 males 27 females | Torque cell | Isometric (Nm) | | Males: Pronated: 11.88 ± 3.75 Supinated: 5.07 ± 1.52 Females: Pronated: 6.04 ± 1.44 Supinated: 3.08 ± 0.85 | Pronated: $d = 1.56$ Supinated: $d = 1.31$ |

Napper et al. (2023), *PeerJ*, DOI 10.7717/peerj.16557

**Table 8** (*continued*)

| Article | Participants (males; females) | Measurement method | Type of contraction | Isokinetic speeds (if applicable) | Outcome values | Effect size (Cohen's *d*) |
|---|---|---|---|---|---|---|
| *Nilsson et al. (2019)* | 14 males (24.3 ± 2.6yrs) 14 females (24.6 ± 2.4yrs) | Force/torque transducer | Isometric (Nm) | | Males: 18–35: 6.1 ± 2.4 36–65: 8.4 ± 2.4 Females: 18–35: 4.1 ± 1.4 36–65: 3.8 ± 0.8 | 18–35: *d* = 0.83 36–65: *d* = 1.92 |
| *Puharic & Bohannon (1993)* | 12 males 12 females | Dynamometer | Isometric (N) | | Males: 38.2 ± 5.4 Females: 20.1 ± 3.1 | *d* = 3.35 |

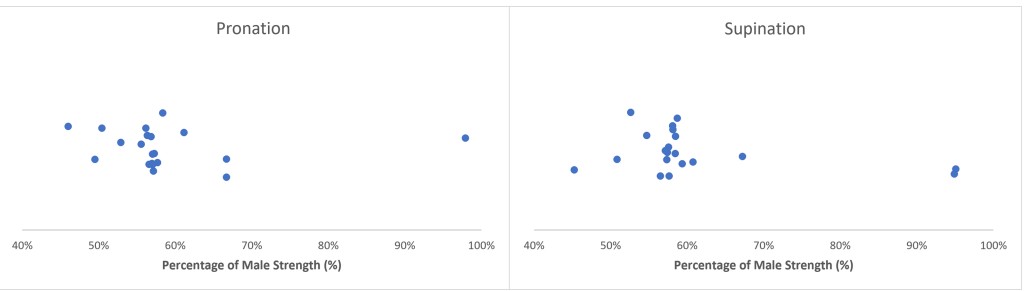

**Figure 4  Normative pronation/supination strengths.** Sex differences in pronation (left) and supination (right) strength with female strength metrics ranging from 46–98% and 45–95% of male strength, respectively. Articles in this review indicate that female pronation/supination strength centralized at ~55–60% of male strength.

## DISCUSSION

The primary aim of this review was to examine sex differences in wrist strength during different movement directions and contraction types. Although sex differences in strength have been heavily investigated, there is a paucity in research regarding sex differences in strength of the distal upper extremity. From this review, sex differences in wrist strength were observed across articles, with females commonly producing lower strength metrics than males with very few exceptions. These differences ranged based on different movement patterns and contraction types, with females generating considerably smaller indexes. Importantly, this review provides ranges, noting that, across all articles, females typically generated ~60–65% of male flexion/extension, ~55–60% of pronation/supination, and ~60–70% ulnar/radial deviation strength. Ergonomists and work task developers should account for decreases in relative strengths that differ based on movement direction and contraction types to improve occupational workplace guidelines.

Female strength centralized around ~55–70% of male strength, but differences existed across movement pairs. Females produced a wider percentage of strength during flexion/extension with a range of 38–144% of male strength; pronation/supination and ulnar/radial deviation only presented ranges of 45–98% and 48–80%, respectively. This is important to note as females generated reduced relative strength during pronation/supination movements, and more so during ulnar/radial deviation. In activities of daily living, individuals commonly adopt the dart-throwers motion (DTM), which is described as moving from extension and slight radial deviation to flexion and slight ulnar deviation (*Anderton et al., 2022*). Many hand tool tasks place workers under high degrees of ulnar/radial deviation, leaving workers with a greater risk of injury during these movements due to decreased strength ability (*Rempel, Keir & Bach, 2008*). These differences in strength across movements are relevant for occupational ergonomists and work task designers, who can leverage these strength ranges for industrial task design; particularly in considerations of increased conservatism for strength requirements when the task requires wrist movement directions outside of flexion/extension.

Isometric exertions were the predominant measurement method for assessing strength in these included articles, where females produced less strength when compared to isokinetic contractions. Females generated ~38–82% of male flexion strength during isometric contractions, and 53–135% during isokinetic. When examining extension movements, females were able to generate ~42–79% and ~49–102% of male strength during isometric and isokinetic contractions, respectively. The large ranges in strength capabilities present the need for conservatism in generating work demands; tasks scaling to male strength will need to consider the range of lower female strength percentages presented in this review. These differences tended to become larger when accounting for bodyweight, where during isokinetic contractions, females produced 144% of male strength in flexion/extension movements (*Mao et al., 2000*). It is possible that normalizing to body mass mitigated the sex effects commonly seen between males and females; additional research under body mass normalized conditions are warranted. Isokinetic contractions and relative strength values were only examined in three articles; this number is insufficient to conclude isokinetic strength generated significant relative strength increases in females, but indicates that further analysis should be conducted on these findings.

This review has identified a large research gap examining sex differences in wrist strength, specifically regarding movement directions, contraction types, and normalized strength metrics. Of the 14 articles examining flexion and extension, only six (*Danneskiold-Samsøe et al., 2009*; *Forthomme et al., 2002*; *Harbo, Brincks & Andersen, 2012*; *Hill et al., 2018*; *Mao et al., 2000*; *Nicholas et al., 1989*) examined isokinetic contractions; four articles examined ulnar and radial deviation, but none completed isokinetic testing, and only one article was found to examine isokinetic contractions during pronation/supination (*Kramer et al., 1993*). Isokinetic contractions are currently underrepresented in this research area; expansion of findings would provide improved insight into sex differences in wrist strength. This review has additionally identified relative strength as a large research gap. Relative strength was only examined during flexion/extension and during isokinetic contractions (*Forthomme et al., 2002*; *Mao et al., 2000*; *Nicholas et al., 1989*), identifying a large existing research gap investigating strength and isometric contractions, as well as ulnar/radial deviation and pronation/supination. At present, it is difficult to conclude whether these strength differences are due to anthropometric and muscular composition differences (*Miller et al., 1993*; *Janssen et al., 2000*; *Bartolomei et al., 2021*) or biases that exist in these articles, but a general trend that females produce lesser strength compared to males is clear.

There are limitations to be considered for this review. Substantial variability in study design was observed in the included articles and prevented statistical interpretation *via* a meta-analysis. The wide array of relative strength percentages outlined in the results may have been generated from confounding outcomes. A minimal number of articles normalized strength metrics to bodyweight, but when this occurred, results conflicted within this review, hampering interpretability (*Forthomme et al., 2002*; *Mao et al., 2000*; *Nicholas et al., 1989*). Further investigations regarding ulnar/radial deviation, pronation/supination and isokinetic contractions are needed to fully understand the complexity of sex differences in wrist strength. Strength differences in injured populations were not examined within this review, where results may differ when including injured

populations where sex differences in wrist strength may become greater or lesser with injury. While additional strength research for the wrist existed, many articles were excluded as they exclusively utilized males or females, preventing opportunities for comparison. Many of the studies within this systematic review have small sample sizes, where additional research with larger participant pools would strengthen our understanding of these differences in strength. Many articles that identified greater female strength did so, only when strength was normalized to body mass; it is possible that this technique alters our interpretation. However, with few studies using this methodology and an inability to definitively re-integrate these values to absolute strengths without knowing the relationships between participant masses and outcome strengths, we identify this as a potential limitation. Future research would benefit from additional studies using this technique, as well as documenting individual participant outcomes to allow research groups to redefine the outcomes as absolute strengths if they so choose.

## CONCLUSIONS

Female strength was observed to be lower than male strength across all wrist motions during isometric contractions for articles included within this review. Some articles demonstrated higher female strength, but these were typically only observed when strength was normalized to participant body weight. Job demands do not often differ based upon anthropometrics and are generally created with the focus on absolute strength, leaving female workers exposed to injury risk. Female workers are expected to work under the same conditions as male workers, therefore requiring them to work at a higher capacity threshold, which may attribute to their increased WMSD risk observed across a magnitude of different occupations (*Tessier-Sherman et al., 2014*; *Islam et al., 2001*). The decreased relative strengths are not at universal levels but differ by movement direction and contraction type; these should be considered by ergonomists and work task designers when examining occupational scenarios. These high demands are associated with wrist/hand pathologies such as carpal tunnel syndrome (CTS) which are already more prevalent in females (*Feng et al., 2021*), indicating the importance of closing this research gap to develop stricter workplace guidelines to aid in injury risk prevention (*Fan et al., 2009*; *Shiri & Viikari-Juntura, 2011*; *Wolf et al., 2010*).

## ACKNOWLEDGEMENTS

The authors would like to thank Ian D. Gordon and the Brock University Library for their expertise with evidence synthesis, database search term development, knowledge translation, and support for this research review.

### Funding

Michael WR Holmes is supported by the Canadian Research Chairs program and an NSERC Discovery Grant (RGPIN 2023-04488). The funders had no role in study design, data collection and analysis, decision to publish, or preparation of the manuscript.

### Grant Disclosures

The following grant information was disclosed by the authors:
Canadian Research Chairs program.
NSERC Discovery Grant: RGPIN 2023-04488.

### Competing Interests

Michael W.R. Holmes is an Academic Editor for PeerJ.

### Author Contributions

- Alexis D. Napper conceived and designed the experiments, performed the experiments, analyzed the data, prepared figures and/or tables, authored or reviewed drafts of the article, and approved the final draft.
- Meera K. Sayal conceived and designed the experiments, performed the experiments, analyzed the data, prepared figures and/or tables, authored or reviewed drafts of the article, and approved the final draft.
- Michael W.R. Holmes conceived and designed the experiments, performed the experiments, prepared figures and/or tables, authored or reviewed drafts of the article, and approved the final draft.
- Alan C. Cudlip conceived and designed the experiments, performed the experiments, analyzed the data, prepared figures and/or tables, authored or reviewed drafts of the article, and approved the final draft.

### Data Availability

This manuscript is a systematic review and did not generate raw data.

### Supplemental Information

Supplemental information for this article can be found online at http://dx.doi.org/10.7717/peerj.16557#supplemental-information.

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
