# Peer review of "Sex differences in wrist strength: a systematic review"

_PeerJ, doi:10.7717/peerj.16557_

## Round 0.1 · original submission · Minor Revisions

Both reviewers require clarifications on a number of points.

Reviewer 1 ·

Basic reporting

This paper presents a systematic review examining the differences in wrist strength between males and females. The writing and reporting of results are both clear and professional. Sufficient context and background have been provided to help understand the need for this research in the field of ergonomics. The presented tables are thorough and present all the relevant data from the systematic review. A few comments for consideration follow below:

1) Lines 76-79: “While differences in strength between sexes has been documented, workplace tasks do not typically scale force or loading requirements to worker anthropometrics or capability, placing disproportionate risk levels on those with decreased capacity”

Though it is true that those with decreased capacity are at a higher risk for overexertion injury, I would argue that many strength-based ergonomics decisions account for this by considering the strength of a weak female (e.g. 25th percentile, or 75% capable) as a design or threshold criterion. This is based on prior epidemiological evidence (Snook, 1978) and reflected in numerous ergonomics approaches (e.g. NIOSH lifting equations - Waters et al, 1993; static strength prediction models - Chaffin et al., 2006). The authors of this paper imply that female strength is rarely considered, which I do not think is a fair characterization. I would recommend that the authors add some discussion about this point, but can use it as further justification for why it is so important to understand discrepancies in strength across sexes.

Experimental design

It seems that the authors conducted a rigorous systematic review that conforms to the PRISMA standards. The authors adequately report on all facets of the review, and it seems that the results turned up a reasonable number of papers directly related to the research question.

As the paper goes on to conduct some initial quantification of strength data, should this paper also be considered as a meta-analysis? It is currently not framed as such, but it perhaps should be.
Lines 205-206: When summary means are provided, are these from a weighted average based on sample size from each study? If not, I think this should be considered to give more weighting to more heavily powered research studies, especially given the challenges and peculiarities of measuring wrist strength.

Validity of the findings

Overall, I believe the results were adequately presented. Comprehensive summary tables from the review provide an excellent resource for researchers to consult.

I was quite confused by the reporting of female strengths above that of males, which seems highly unlikely to be a true result based on the biomechanical and physiological differences between males and females, on average. In the discussion section, the authors describe that these scenarios (female strength >100% of male strength) occurred when the strengths were normalized to body weight. I don’t think it makes sense to include both absolute and normalized to BW strength ratios in the same metric, as they are very different representations of strength. If the paper reports the average mass of males and females, can you perhaps calculate what the non-normalized (i.e. absolute) strength would be, at least approximately, for the papers that only provide normalized data?

Additional comments

No additional comments.

Cite this review as

Reviewer 2 ·

Basic reporting

The authors conducted a systematic literature search and review to examine sex differences in write strength during different movement directions and contraction types. However, the analysis is descriptive and a summary of exiting evidence. It’s not clear to me what the contribution of the manuscript is to the literature.

Experimental design

In the literature search section, the authors did not clearly specify the study type, study population and study outcomes of interest to be included in the search and the following analysis. Different study outcomes within different types of studies have different clinical implications. It’s important to clearly define these before conducting the search. Please clarify and elaborate.

Validity of the findings

Some of the studies the authors included in the meta-analysis only include a relatively small number of patients (e.g., less than 30 patients). Is the evidence summarized in the manuscript sufficient to support the conclusion?

Cite this review as

---

## Round 0.2 · accepted · Accept

I have assessed the authors' responses. In my opinion, the manuscript can be accepted without further consultation with the reviewers.